# The Outcomes of Patients with Haemoglobin Disorders in Cyprus: A Joined Report of the Thalassaemia International Federation and the Nicosia and Paphos Thalassaemia Centres (State Health Services Organisation)

**Michael Angastiniotis [1,\*], Soteroula Christou [2], Annita Kolnakou [3], Evangelia Pangalou [2], Irene Savvidou [2], Dimitrios Farmakis [4] and Androulla Eleftheriou [1]**

[1] Thalassaemia International Federation, Nicosia 2007, Cyprus
[2] Cyprus Thalassaemia Centre, Strovolos 1474, Cyprus
[3] Paphos Thalassaemia Centre, Paphos General Hospital, Paphos 8026, Cyprus
[4] Department of Physiology, University of Cyprus Medical School, Nicosia 1678, Cyprus
[\*] Correspondence: michael.angastiniotis@thalassaemia.org.cy

**Abstract:** Haemoglobinopathies, including thalassaemias and sickle-cell syndromes, are demanding, lifelong conditions that pose a significant burden to patients, families, and healthcare systems. Despite the therapeutic advances and the resulting improvements in prognosis accomplished in past decades, these patients still face important challenges, including suboptimal access to quality care in areas with developing economies, changing epidemiology due to massive migration flows, an evolving clinical spectrum due to ageing in well-treated patients, and limited access to novel high-cost therapies. We herein describe the organization of healthcare services for haemoglobinopathies in Cyprus—with particular focus on beta-thalassaemia, the most prevalent condition in this region—along with selected patient outcomes. This report aims at underscoring the fact that nationally funded and well-coordinated prevention and care programmes for chronic and complex conditions, such as haemoglobinopathies, with active involvement from patient organizations lead to effective disease control and excellent outcomes in survival, quality of life, social adaptation, and public health savings, and allow timely and effective responses to emerging crises, such as the COVID-19 pandemic. The Cyprus paradigm could therefore serve as a blueprint for the organization or adaptation of haemoglobinopathy programs in other countries since these disorders are still widely occurring.

**Keywords:** haemoglobin disorders; haemoglobinopathies; thalassaemia; transfusion; iron overload; public health

## 1. Introduction

Haemoglobin disorders are inherited haemoglobin (Hb) disorders resulting from mutations of the beta (β) or alpha (α) globin chain genes. The main clinically significant syndromes include β-thalassaemia, sickle cell disease (SCD) syndromes, HbH disease (a type of α-thalassaemia), and Hb E/β-thalassaemia [1]. These disorders, originally prevalent in malaria-endemic, or previously malaria-endemic, areas of the world, including southeast Asia, the Middle East, southern Mediterranean countries, and northern and central Africa, today occur widely across the world due to mass migrations of populations from high prevalence areas. Still, however, approximately 80% of children born with a haemoglobin disorder each year live in developing countries or areas with developing economies [2]. A broad spectrum of clinical severity characterizes these disorders based on (i) the various mutations or mutation combinations that cause them, (ii) the vast heterogeneity of the quality of provided care, (iii) the challenges related to adherence to treatment, (iv) socio-economic status, and (v) migration conditions.

Today, after many decades of clinical progress, effective primary and secondary prevention and clinical management implemented in the context of nationally coordinated disease-specific strategies have resulted in dramatically improved survival and quality of life for patients with these disorders [3]. The rates of morbidity and mortality, however, are—as described above— entirely dependent on the quality of medical, social, and other care provided nationally. The development of serious, chronic, and life-threatening comorbidities becomes more pronounced in fragile health systems, particularly those without universal coverage, in which patients and families also bear the financial strain of treatment for these lifelong, highly demanding diseases. In such environments, equity of care is severely compromised as a result of the plethora of other public health concerns, including—to date in many regions of the developing world—communicable and, more commonly, non-communicable diseases, such as diabetes, heart disease, and cancer; societal issues, such as poverty, political instability, health, education, and literacy; and extraordinary and unexpected circumstances, including serious infectious outbreaks, such as SARS, MERS, Ebola, and currently the COVID-19 pandemic.

Complications in β-thalassaemia are mainly due to iron overload affecting vital organs, while in SCD, the complications are mostly related to vaso-occlusive episodes. These complications are multi-organ in both syndromes and so require close monitoring and the involvement of various medical specialities. The requirement of lifelong, regular blood transfusions in many patients—not only confined to transfusion-dependent thalassaemia—and the need for specially processed blood for these patients, has resulted in the need for dedicated day-transfusion centres, separate from day care units for malignancies. The concept of coordinated multi-disciplinary care; detailed, comprehensive monitoring of organ function and damage; and the effectiveness of the different therapies, particularly iron chelation, has additionally created the need for the establishment of reference/expert centres.

The complexity of services that are necessary for patient survival and wellbeing puts a significant burden on healthcare systems as well as on patients and their families. Any reduction in quantity or quality of services brings with it a reduction in desired outcomes [4]. This in turn creates the need for more usage of inpatient and intensive care services and raises the cost of care even further or results in premature death.

In high-prevalence countries, such as Cyprus, these needs were identified as early as the 1960s, when knowledge of treatment was rudimentary and was only focused on systematizing blood transfusion to correct anaemia and to suppress endogenous ineffective erythropoiesis. Blood donation has since become a major issue because voluntary donation was not the norm and families were obliged to find donors for their children. By the early 1970s, increasing use of iron chelation to reduce iron loading resulting from frequent transfusions brought to light the complexity and expense of the services, with iron chelation being an additional strain on the families' budgets at the time. In this evolving picture, public health authorities in some—albeit very few—countries, including Cyprus, responded in two ways. An attempt was made to reduce the number of affected new births, while at the same time increasing the support for families in providing for their children [5,6], until all costs related to the necessary services provided to patients were included, with the support and demand of the National Patients' Associations in 1978, in a national programme fully funded by the government.

## 2. Haemoglobin Disorders in Cyprus

Cyprus, an island state in the eastern Mediterranean of approximately 1.2 million inhabitants, has one of the highest carrier rates for β-thalassaemia in the world, at 12–15% [7]. The carrier rate for α-thalassaemia is 20%, whereas SCD carriers are significantly less, at 0.2% of the population. Five mutations account for more than 95% of all β-thalassaemia alleles in the patient population [8]. The most common β-globin gene mutation is IVS I-110 (G > A), with a percentage of 74–80%, followed by alleles IVS I-6 (T > C) and IVS I-1 (G > A), with frequencies of 12.4% and 5.1%, respectively. This was confirmed early on to be true for the whole population of the island [9]. In addition, there are various haemoglobin variants

in low prevalence, such as Hb D, Hb Lepore, Hb Setif, Knossos, and others [10]. All patients in Cyprus are characterised by their full genotype of both β and α thalassaemia mutations.

Historically, as previously mentioned, Cyprus took measures early. Screening to identify carriers was initiated in 1972, and by 1976 Cyprus was sending 'at-risk' carrier couples to the UK for prenatal diagnosis until such a service was established locally in 1980 [11]. A patient-support organisation called the Pancyprian Antianemic Society (PAS), formed in 1970, advocated effectively for—and, by 1978, succeeded in obtaining—full health coverage, for all patients, of all treatment services through the government budget. Additionally, in 1979 a new, government-supported NGO was created to promote blood donation; this was driven by volunteers, mostly parents and doctors dealing with thalassaemia. This initiative was successful in sensitising the community to voluntary non-remunerated blood donation, and quickly resulted in the elimination of, or at least a significant reduction in, the need for any replacement or paid system, not only for thalassaemia but also for any other medical indication for blood transfusion.

The establishment of PAS, the patient-support group, originally formed by parents of very young patients, has played a leading role in all developments in the country, both in the effectiveness of the prevention programme, since it aimed to save resources for their children, but also in the development of clinical services. The combined efforts of PAS, the medical team, the state, and the Church brought about the initial creation of the thalassaemia centre in the capital city of Nicosia. This was officially opened in 1983 as a separate building attached to the paediatric hospital (patients in those days were mostly in the paediatric age group). The Centre, on account of its gathered expertise and the success of its programme, was assigned in 1986 as a WHO Collaborating Centre. Two wings comprise the centre: one hosts the screening and diagnosis laboratories, and the other operates as a day care unit for the clinical care of patients, with consulting rooms and a day-transfusion centre. This was followed some years later with the creation of separate thalassaemia units in the main peripheral hospitals to cover the needs of the Larnaca/Famagusta, Limassol, and Paphos areas.

Laboratory technology was centralised in the Nicosia clinic even after the development of the other three regional clinics. This included one fully equipped, state-of-art haematology laboratory, part of the public health services of the Cyprus Ministry of Health. An official collaboration with the Cyprus Institute of Neurology and Genetics was established in the early 1990s, and provided state-of-art molecular services both for diagnosis, screening, and prenatal testing, as well as, very importantly, for research.

In this paper we describe the current state of services for the control of haemoglobin disorders in Cyprus. The purpose of this description is to demonstrate that national planning and coordination with fully funded services, along with active and productive community involvement, are essential components in the achievement of good outcomes for patients and public health savings, which can in turn address effectively serious threats including infectious diseases outbreaks, such as the COVID-19 pandemic.

## 3. Description of Services

The prevention and treatment of thalassaemia and SCD are governed by a disease-specific programme, implemented in 1972 by the national health authorities of Cyprus [11]. The programme recognized the necessity of dedicated services to be provided for these complex and chronic diseases, thus establishing distinct thalassaemia clinics in four regions of the country (Nicosia, Limassol, Larnaca/Famagusta, and Paphos), administered by public hospitals. These clinics are, to this day, responsible for the treatment and multidisciplinary care of all patients (thalassaemia and SCD) on the island. They constitute day care centres for consultation, monitoring, transfusions, and out-patient care for both adults and (with the support of paediatricians) paediatric patients, as well as engaging at the same time with a significant number of medical, paramedical, and other personnel in the context of the multidisciplinary care needed. The doctor/patient ratio for regularly attending beta thalassaemia and sickle cell patients is 1:89, while the nurse/patient ratio is 1:30.

The four thalassaemia clinics to-date function as independent entities to each other, but nonetheless, due to the close collaboration of the medical personnel, the procedures, pathways, and parameters of treatment protocols are near-identical, as they all follow the relevant TIF's mainly international guidelines for standards of care and the management of thalassaemia [12]. All thalassemia patients receive treatment at one of the four large thalassemia centres in the country, with the exception of two cases who attend private clinics.

To achieve multidisciplinary care and share knowledge and expertise with each other, the thalassaemia clinics in Cyprus have engaged, either separately or jointly, in a number of collaborations at the national, European, and international level.

*Blood transfusion:* Concerning the provision of blood, the National Blood Establishment in Nicosia and the hospital blood banks in each city are responsible for the collection, processing, and storage of blood and its components, as well as for their distribution to thalassaemia centres across the country, with each blood bank serving the needs of the patients of the corresponding hospitals it belongs to. It is noted that Cyprus is one of the few countries in the world that is self-sufficient in blood and blood product requirements, stemming from a historical tradition, as mentioned previously, of voluntary non-remunerated blood donation, promoted by the special Blood Donation Coordinating Committee established in the 1980s, initially for the needs of patients with transfusion-dependent thalassaemia (and subsequently expanded to cover all needs of the Cyprus population for blood transfusions). Following processing and pre-storage filtration, packed red blood cell units are prepared for transfusion into patients; most transfusion-dependent patients receive blood every 6–13 days, according to treating physicians' assessments of individual patient needs. For the majority of patients, pre-transfusion Hb level is maintained at 10 g/dL or higher; each unit is screened for hepatitis B and C, human immunodeficiency viruses 1 and 2, and syphilis (as well as cytomegalovirus for selected patients). Screening of the blood products for potential infections is performed using nucleic acid tests.

*Iron chelation* is performed by the use of all three officially authorised chelating agents, in doses and combinations tailored to each individual patient's needs as these are dictated by regular monitoring, both biochemical and imaging, according to international guidelines [12].

*Multidisciplinary collaboration* is maintained with specialists both in public hospitals as well as in private practice, including (but not limited to) cardiologists, endocrinologists, and hepatologists for the multi-disciplinary care of all patients across the island.

*Magnetic resonance imaging services* for the estimation of iron load in the heart and liver are offered by both the hospital radiology departments and private centres, and in both cases are fully supported by a governmental budget. Each patient is always measured in the same centre to ensure consistency, and an attempt is made every year to appropriately calibrate the T2* software to measure as accurately and as reproducibly as possible cardiac and liver iron content.

*All laboratory support* for patient diagnosis and monitoring is provided by the hospital laboratories, including specialised endocrinology, virology, and other tests.

*Electronic patient records* are kept only in the Nicosia clinic at the time of writing, while a new updated disease-specific EMR, developed by the Thalassaemia International Federation, is due to be adopted by all clinics by December 2022.

*Professional psychological support* was nearly the only service not provided by the Ministry of Health, but was periodically supported by the local patient support associations.

The *day care clinics* have a 12 h schedule (7 am–7 or 8 pm) to allow patients to attend after school or employment hours, facilitating their full social/professional integration. However, there is no weekend service.

At the time of writing, health services in Cyprus are undergoing major reforms uniting public and private sectors under one health insurance scheme, greatly narrowing the public/private divide and providing, through taxation, equal access of every citizen to healthcare services.

HSCT is not available in the country. Young patients, mainly, are referred to centres in Italy, Greece, or Germany sponsored by government funds. So far 32 cases have been completed.

## 4. Patient Numbers

Currently, there are 1244 patients with red cell disorders followed at four thalassaemia centres on the island, including 594 with β-thalassaemia major, 63 with β-thalassaemia intermedia, 475 with HbH disease, and 56 with SCD. These are geographically distributed across four regions of the island (Nicosia, Limassol, Larnaca/Famagusta, and Paphos, Table 1). There are 657 patients with β-thalassaemia syndromes, of whom 90.4% are transfusion-dependent and 9.6% are not transfusion-dependent. Of the 56 cases of sickle cell disease, only one is HbSS homozygote, with the rest being HbS/β-thalassaemia. Not all HbH disease patients are on regular follow-up, since many come for irregular check-ups due to the mild phenotype. Of the 'other anaemias', the majority are hereditary spherocytosis, with a few cases of dyserythrpoietic anaemia and Blackfan Diamond syndrome.

**Table 1.** Distribution of patients with haemoglobin disorders in Cyprus in the various treatment centres according to the diagnostic group.

| Clinic | β-Thal. Major | β-Thal. Intermedia | HbH Disease | Sickle/ β-Thal. | Other | BMT | Total |
|---|---|---|---|---|---|---|---|
| Nicosia | 244 | 42 | 258 | 28 | 11 | 10 | 593 |
| Limassol | 174 | 9 | 94 | 12 | 11 | 5 | 305 |
| Larnaca | 135 | 5 | 102 | 10 | 1 | 16 | 269 |
| Paphos | 41 | 7 | 21 | 6 | 0 | 1 | 76 |
| Total | 594 | 63 | 475 | 56 | 23 | 32 | 1243 |

Thal.: thalassaemia; HbH: haemoglobin H disease; BMT: bone marrow transplantation.

## 5. The Report

This report describes some parameters related to patient outcomes deriving from the whole patient population (657), but focuses more comprehensively on the data from patients with β-thalassaemia syndromes in two regions, Nicosia and Paphos. Data were extracted from the thalassaemia registry of the two thalassaemia clinics, the central Nicosia Thalassaemia Centre and the Paphos Thalassaemia Centre of Paphos General Hospital, with a combined current β-thalassaemia population of 331 patients, representing 47.6% of the total population in Cyprus. Results of magnetic resonance imaging-derived cardiac iron load were available in 96.5% of patients, and results of liver iron concentration (LIC) in 85.6%. Survival data and complication rates are derived from the national thalassaemia registry, which currently includes 657 living patients.

## 6. Survival

Among 657 patients with β-thalassaemia, only 72 (11%) are aged <18 years (10.6% with TM and 14.3% with TI); the current age distribution curve is provided in Figure 1. The mean age of patients 42.3 years (median 44 years), with a gradual reduction due to patient loss in the 6th decade. In the young age groups, the small number of patients reflects the limitation of new births, since at-risk couples are screened mainly in the premarital period and offered choices concerning their reproductive life [5]. Improved survival rates seem to be related to improved control of iron overload (see Tables 2 and 3), resulting in a reduction in organ damage over time. More detailed survival data were published in previous years [13,14].

In an updated study of 538 TDT patients born after 1 January 1960 and validated up to 31 December 2018, the median age was 40.9 years [15]. There were 95 deaths and the causes were classified as follows: 47.4% due to heart complications, 14.7% due to infections, 9.5% due to liver disease (including hepatocellular carcinoma), and 8.4% due to other malignancies. The cardiac deaths have shown a downward trend over the years but there

was an increase in hepatic deaths, infections, and malignancies. Estimated survival at the age of 40 years was 89%.

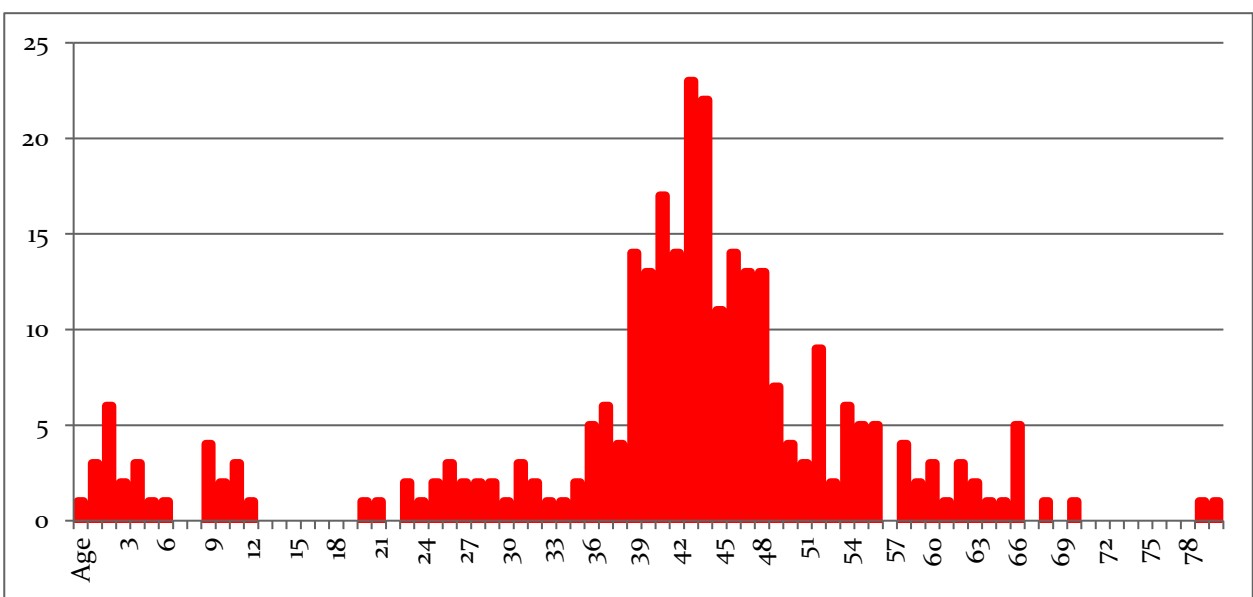

**Figure 1.** Age distribution of patients with β-thalassaemia in Cyprus.

**Table 2.** Prevalence of complications in a Cypriot cohort of β-thalassaemia patients.

| Complication | Prevalence |
|---|---|
| N | 647 |
| Median age, years | 43 |
| Bone disease | 218 (33.7%) |
| Hypogonadism | 184 (28.4%) |
| Diabetes | 95 (14.7%) |
| Hypothyroidism | 84 (13%) |
| Heart disease | 57 (8.8%) |
| Thrombotic episodes | 38 (5.9%) |
| Chronic hepatitis C | 18 (2.8%) |
| Liver cirrhosis | 10 (1.5%) |
| Hypoparathyroidism | 8 (1.2%) |
| Chronic hepatitis B | 5 (0.8%) |

**Table 3.** Iron-load status in patients with thalassaemia in Cyprus; serum ferritin, cardiac iron load, and liver iron concentration (LIC) according to transfusion status.

| | Regularly Transfused N (%) | Not Regularly Transfused N (%) | Total Patients N (%) |
|---|---|---|---|
| Serum Ferritin (ng/dL) | | | |
| <1000 | 136 (48.9%) | 40 (75.5%) | 176 (53.2%) |
| 1–2500 | 72 (25.9%) | 5 (9.4%) | 77 (23.3%) |
| >2500 | 70 (25.2%) | 8 (15.1%) | 78 (23.5%) |
| Totals | 278 (100%) | 53 (100%) | 331 (100%) |
| Cardiac T2* (ms) | | | |
| >20 ms | 228 (85.4%) | 35 (100%) | 263 (87%) |
| 10–19 ms | 27 (10.1%) | 0 | 27 (9%) |
| <10 ms | 12 (4.5%) | 0 | 12 (4%) |
| Totals | 267 (100%) | 35 (100%) | 302 (100%) |

**Table 3.** *Cont.*

|  | Regularly Transfused N (%) | Not Regularly Transfused N (%) | Total Patients N (%) |
|---|---|---|---|
| LIC (mg/g dry weight) |  |  |  |
| <3 | 121 (51%) | 19 (61.3%) | 140 (52.2%) |
| 3–7 | 53 (22.5%) | 4 (12.9%) | 57 (21.3%) |
| 7–15 | 31 (13%) | 5 (16.1%) | 36 (13.4%) |
| >15 | 32 (13.5%) | 3 (9.7%) | 35 (13.1%) |
| Totals | 237 (100%) | 31 (100%) | 268 (100%) |

Survival of patients with β-thalassaemia in Cyprus according to birth cohort is given in Figure 2.

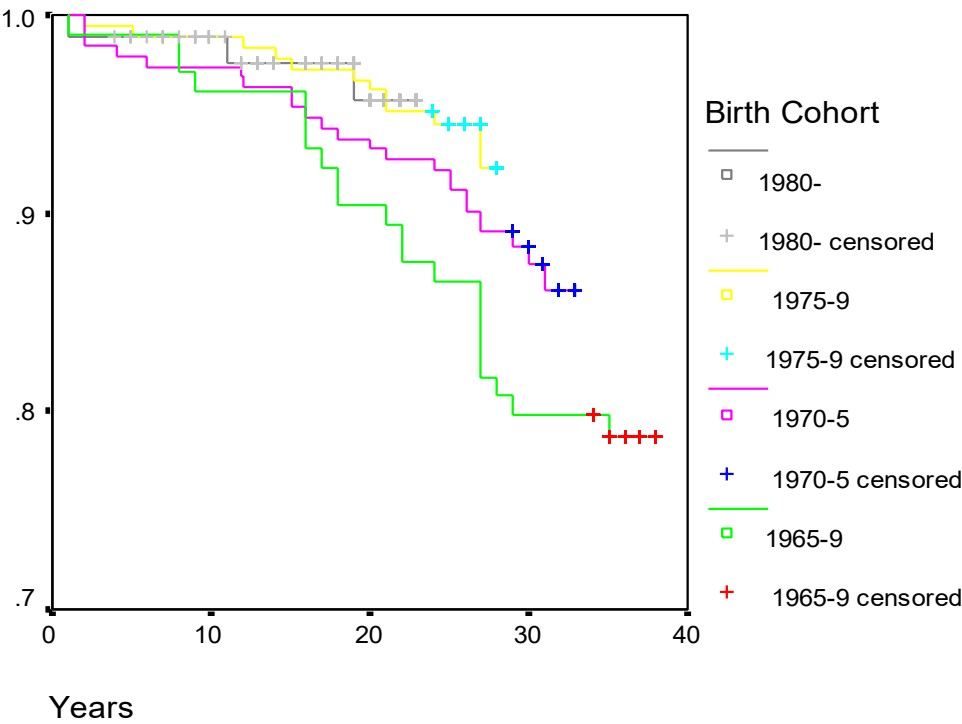

**Figure 2.** Survival of patients with β-thalassaemia in Cyprus according to birth cohort.

## 7. Complications

Complication rate is an important indicator of clinical outcomes and the effectiveness of clinical care over time. The recorded complications in a survey of 647 thalassaemia patients (including both transfusion-dependent and non-transfusion-dependent patients) with a mean age of 43 years, derived from the national survey, are summarized in Table 2. These are expected complications related to iron overload and other factors, which reflect the age of this population of thalassaemia patients, in which 88% are over the age of 18 years. The most frequent complications were bone disease, mainly osteoporosis, followed by hypogonadism, diabetes, hypothyroidism, and heart disease. A comparison of complication rates with a Greek β-thalassaemia cohort [16] is provided in Supplementary Table S1.

A more detailed analysis was undertaken from the patient records of 331 patients, selected from two thalassaemia clinics in Cyprus (Nicosia and Paphos), and updated to 1 July 2020. Of these, 35 (10.6%) are under the age of 18 years, and 124 have been splenectomised (37.5%). However, no patients under the age of 18 years have been splenectomised, so 41.9% of patients over the age of 18 years have been splenectomised. Of 296 patients over

the age of 18 years, 259 are regularly transfused: of these, 87 are splenectomised (33.6%). Of 53 non-transfusion-dependent patients, 37 are splenectomised (69.8%).

Table 3 outlines the iron load status of patients with β-thalassaemia in terms of serum ferritin concentration, cardiac iron, and LIC. The majority of patients have serum ferritin, cardiac iron, and LIC levels ranging well within the safety level, with better control of cardiac as opposed to liver iron. This good control in the majority of patients is due to regular monitoring, including MRI iron measurements and, when needed, rescue therapy by regimens of a combination of iron chelating agents. A comparison of iron-load status among Cypriot patients and patients originating from other parts of the world, including Egypt, Greece, Indonesia, and Australia, is provided in Supplementary Tables S2 and S3 [17–20].

In an adult and aging thalassaemia population, malignancies are becoming more common. Until the end of the 20th century, these were regarded as isolated and even rare events, while in more recent years this group of diseases is increasingly contributing to both morbidity and mortality [21–24]. It is expected that this current thalassaemia group in Cyprus will also share the same experience. In a population of 331 patients, 26 cancer cases (7.9%) occurred after the year 2000. The mortality rate is high, accounting for 53.8% in this cohort (14 deaths), and is increasingly contributing to the causes of death. As in other studies, hepatocellular carcinoma is more frequent in non-transfusion-dependent thalassaemia, but overall there is an increasing variety in the types of cancer. These data require further analysis, and a detailed study is now in preparation.

## 8. Social Aspects

Improved patient care must result in a good quality of life and social integration of these chronically affected patients. This is a subject deserving of a separate in-depth study, but from the patients' records we have gathered limited indicators to demonstrate a level of success on the social level. Such indicators include education, employment, marriage, and having children. These are indicators of patient confidence in life and in their treatment, and of a desire to fulfil much of life's normal expectations. Education and employment status of β-thalassaemia patients in Cyprus is summarized in Table 4 and compared with cohorts originating from Greece and Iran in Table S4 [25,26]. The majority of patients are employed or retired, while the unemployment rate is 8.2%. In a study from Greece [26], a group of 109 adult thalassaemia patients had a similar rate of paid employment (77%), while a higher percentage, 23%, were unemployed, consistent with the country's general unemployment rate. Overall, there are very few recent publications dealing with the issue of employment in thalassaemia patients and the comparisons are difficult due to very different social and cultural environments.

Marital status and child bearing of β-thalassaemia patients in Cyprus are summarized in Table 5. Cultural similarities in terms of marriage practices can be compared between Cyprus and Greece or Italy. In the Greek study [26], 36.7% of patients were married, very similar to the Cypriot group, while 52.9% were unmarried or divorced (in the Cypriot group these categories account for 49.3%, and if cohabitation is added then 51.5% are accounted for, almost identical proportions). In an international study conducted by the ICET-A group, which studies mainly endocrinological complications of thalassaemia [27], in a thalassaemia population numbering 966 (consisting of both regularly transfused and non-transfusion-dependent patients from both Europe and Asia, including Cyprus), only 24.8% were married or cohabiting; in both the current study and in the ICET-A study, among adult Cypriot patients, over 50% had fulfilled this life ambition. In terms of child bearing, assisted reproduction was necessary in many cases, and twins and triplets have often been the result. One maternal death from pulmonary embolism has been recorded but otherwise most pregnancies were safe. It is not intended here to analyse the outcomes of pregnancies in detail, even though a more recent review is overdue [28,29].

**Table 4.** Levels of education and employment in patients with thalassaemia in Cyprus according to transfusion status.

| | Regularly Transfused N (%) | Not Regularly Transfused N (%) | Total N (%) |
|---|---|---|---|
| Level of education | | | |
| None (infants) | 3 (1.2%) | 2 (5.3%) | 5 (1.7%) |
| Primary | 15 (6%) | 11 (28.9%) | 26 (9%) |
| Secondary | 160 (63.7%) | 19 (50%) | 179 (62%) |
| Tertiary | 73 (29.1%) | 6 (15.8%) | 79 (27.3%) |
| Total | 251 (86.9%) | 38 (13.1%) | 289 (100%) |
| Level of employment | | | |
| Working full-time | 189 (74.1%) | 19 (52.8%) | 208 (71.5%) |
| Working part-time | 3 (1.2%) | 1 (2.8%) | 4 (1.4%) |
| Unemployed | 21 (8.2%) | 3 (8.3%) | 24 (8.2%) |
| Retired | 10 (3.9%) | 9 (25%) | 19 (6.5%) |
| "Disabled" | 32 (12.6%) | 4 (11.1%) | 36 (12.4%) |
| Total | 255 (100%) | 36 (100%) | 291 (100%) |

Retired: the patient has completed the years of service (by Cyprus law this is 30 years) and is now on a pension; "Disabled": complications of thalassaemia or other adverse health events have forced the patient to withdraw from workforce and accept a disability pension.

**Table 5.** Marital status and child bearing in patients with thalassaemia in Cyprus according to transfusion status.

| | Regularly Transfused N (%) | Not Regularly Transfused N (%) | Total N (%) |
|---|---|---|---|
| Marital status | | | |
| Single | 96 (35.6%) | 16 (33.3%) | 112 (35.2%) |
| Married | 131 (48.5%) | 30 (62.5%) | 161 (50.6%) |
| Cohabiting | 6 (2.2%) | 0 | 6 (1.9%) |
| Divorced | 37 (13.7%) | 2 (4.2% | 39 (12.3%) |
| Total | 270 (100%) | 48 (100%) | 318 (100%) |
| Child bearing (among 296 adults) | | | |
| Pregnancies | 165 | 26 | 191 |
| Children born | 178 | 26 | 204 |

Achieving life goals such as marriage, parenthood, and employment are related to quality of medical care, prolongation of life, and reduced complication rates. However, other social determinants, such as societal and employer acceptance and the degree of support from family and friends, play a role in this urge for a 'normal' life [30,31].

Psychosocial support is provided by professional psychologists, financed by the patient support association. Experience with health-related quality of life tools [32] indicate good results in 90% of patients. Depression, anxiety, and other negative effects of poor support and maladaptive coping are variable both in prevalence and over time in each patient's life. Different results are obtained in different environments [33–35]. This is an area requiring further and deeper investigation, which is now underway in an ongoing project in Cyprus and Greece, using qualitative methods to elicit patient responses [36].

## 9. Reaction of Services to COVID-19 Challenge

The new coronavirus (SARS-CoV-2) infection presents particular challenges and risks to patients with haemoglobin disorders, who may be more likely to be at increased risk of complications from the new coronavirus disease (COVID-19), even though clinical data at the time of writing are very limited [28]. The clinical manifestations of COVID-19 are mainly respiratory, varying from asymptomatic to flu-like symptoms, but in vulnerable individuals they may progress to severe respiratory complications. Although haemoglobin disorders are not generally associated with respiratory conditions, due to the multi-organ

nature of thalassaemia and sickle cell disease, along with complications of the heart, lungs, and the immune system, a possible infection with SARS-CoV-2 may nonetheless trigger serious life-threatening complications (e.g., acute chest syndrome in SCD patients).

The COVID-19 pandemic provided a further challenge to the organisational structure and the collaboration between clinics, since those with haemoglobin disorders were very early on officially recognised as a high-risk group in the context of national precautionary measures. Since in Cyprus each of the five thalassaemia clinics is attached to one of the public hospitals and all together cover the needs of patients across the island—Nicosia, Larnaca/Famagusta, Limassol, and Paphos—they all followed almost identical precautionary measures as those mandated by the WHO, the EU, TIF, and the national health authorities. However, additional measures were adopted to ensure, to the maximum level possible, the safety of both patients and healthcare personnel and these included the following:

For any medical problem, patients were requested to contact the treating physician via telephone to report their symptoms. The treating physician then advised them accordingly on their next actions.

All thalassaemia clinics created an safe entry/exit pathway by allowing the passage of people only through one door, as well as closing off corridors and interconnecting paths with other hospital departments and wards.

A triage area was set up outside the entrance point, where all persons (patients and personnel) entering the thalassaemia clinic were obliged to respond to a set of questions (e.g., recent travel, presence of respiratory symptoms, fever, etc.) and their temperature was taken.

An isolation room was designated for any suspect cases, including for transfusion.

For transfusion procedures: Patients and personnel were divided into separate and distinct cohorts, which accessed the clinic only by appointment to avoid any overlap between cohorts in terms of time. The size of the cohorts depended on the maximum area capacity of each thalassaemia clinic and, more specifically, of the transfusion areas. This enabled adherence to the distancing recommendations.

For securing blood adequacy during the lockdown, the Cyprus Blood Establishment took a pivotal role both in organizing donor-empowering campaigns and in securing the safety of blood and blood donors themselves. Monitoring blood reserves at safe levels during the COVID-19 pandemic in Cyprus remained a priority. During the national lockdown, blood donors were excluded from the once-a-day restriction on leaving the house and were given the freedom to access the donation centres.

The adoption of the above measures contributed to the protection of patients, resulting in no patients with any haemoglobin disorder being infected with COVID-19, with the exception of one TM patient who arrived from abroad carrying the virus. This patient had a moderate clinical picture and was hospitalised for 4 days, followed by a full recovery. In addition, one medical specialist of the multidisciplinary team in Nicosia and one nurse from the Paphos clinic were infected, but thorough testing of their contacts for SARS-CoV-2-RNA demonstrated that they had not transmitted the infection. Two thalassaemia patients with flu-like symptoms were isolated as a precaution, but repeatedly tested negative.

## 10. Discussion

The description of services and outcomes provided in this report was made to demonstrate that services for complex chronic disorders require nationally coordinated organisation, with both medical and community involvement. Any programme can be severely challenged in an emergency situation such as that caused by the pandemic, which involved strict measures of lockdown, social isolation, and possible supply shortages. An overview of general patient services and outcomes, together with the response of the same services to the recent pandemic, may seem unrelated. What brings the two issues under the same title is the requirement for a combined response by health authorities, medical staff, and patient organisations to promote the best possible outcomes. For COVID-19, no doubt many centres across the world have taken similar measures; the difference here is that

there was a consensus agreement and a fully coordinated action between the centres, even though there was no specific relevant directive from health authorities.

For Cyprus, thalassaemia has been a major public health issue since the 1960s, when patient survival began to improve, increasing the demand on blood donation and for essential, mainly iron chelation, drugs. As medical care continued to improve over time, patient survival increased even further, and introduced more complications, mainly due to iron overload and toxicity, which in turn increased demands on health care resources, as thalassaemia evolved to a multi-organ disease. The limitation of new affected births, which resulted from an effective prevention policy initiated in the early 1970s, helped greatly but did not eliminate the need to increase services for a patient population which was increasing in age. The response of the population, especially from at-risk couples who were detected by premarital screening, has been positive; this is mainly due to a small population (around 1 million), with a high rate of literacy and knowledge of thalassaemia, due to the high prevalence.

The response of health services in Cyprus, as in most developed health care systems, was one of willingly planning to meet the increasing demand and expenses. Recognising the need to provide universal health coverage to meet the needs of a demanding chronic disease has been a social solidarity measure that has relieved families of an unbearable responsibility. Coupled with health system planning and adoption of medical advances in the field, this has allowed many patients to survive and achieve educational, marital, and other aims. When challenged with unexpected emergencies, including not only the current pandemic but also political upheaval over the years, the same solidarity of social partners has allowed for minimal disturbance of patient care.

Patient outcomes are not perfect under current conventional clinical management protocols, but they have indeed significantly improved over the years. The global community of patients affected by haemoglobin disorders has been waiting for promising new approaches to achieve even better results and, very importantly, improved quality of life or even cures for these lifelong conditions. In recent times, such new approaches have indeed been authorised by regulatory authorities and are today on our doorstep. This brings to the forefront a new challenge, which is the access to and affordability of these new technologically demanding and costly treatments. The kind of solidarity and collaboration of stakeholders that Cyprus has experienced in the past needs to be rekindled and strengthened to meet these new challenges.

The Cyprus thalassaemia services are not perfect, and certainly many gaps still exist. One, for example, is that the Ministry has not to date provided a career structure for the medical staff which are today considered, not unjustly, to be dead-end jobs. The national health authorities have brought barriers, on account of their huge bureaucracy, to clinical research instead of actively encouraging it. They have allowed a WHO reference centre, a status given to Nicosia Thalassaemia Centre by the WHO in the 1980s, to be lost. To respond to these and other deficiencies, a National Thalassaemia Committee was established in 2018 with a consultative role to the Cyprus Minister of Health. Members of this Committee include clinicians, pharmacists, blood establishment professionals, patient representatives (PAS), the Institute of Neurology and Genetics, and representatives from the Thalassaemia International Federation. It is expected that its suggestions and recommendations will be taken up and adopted by the decision makers at the Cyprus Ministry of Health.

The description within this report is meant to demonstrate that services such as those that exist in Cyprus are needed in every country where chronic conditions such as haemoglobin disorders are prevalent. Economic and social factors, which may seem to be insurmountable barriers to acceptable service provision in high prevalence countries, can only be met through partnerships between government, communities of health care professionals and patients, and the community at large, such as the partnerships that Cyprus has adopted throughout its history of addressing and controlling thalassaemia on the island.

Data from publications from other centres in other countries are not strictly comparable. They are used here as a broad measure of where the Cyprus services stand and as a reference for the recognition of weaknesses and strengths, providing direction for future planning. Centres of excellence exist even in countries with a relatively low Human Development Index (HDI). Some examples (apart from countries in the Western world, such as the UK, France, Italy, and Greece) include countries such as the UAE, Lebanon, Malaysia, Indonesia, and Thailand. The Thalassaemia International Federation is developing a programme and a system of recognition for such centres, based on disease-specific criteria, and on its work at the EU-level in the context of the ENERCA project, promoting the proliferation of reference centres and ultimately leading up to a system of accreditation. This is a recognition of the fact that quality of care results in a better quality of life and better outcomes in chronic and inherited diseases. However, countries such as Cyprus, even with nationally coordinated services, should always be on the alert that results can be even better if the needs of the new generation of patients are better recognised and met in a timely fashion. New curative approaches are on the way, and some are with us already, but they will not eliminate the need for such quality services, and access to these new therapies will present a huge challenge which can only be met through a nationally, and even globally, co-ordinated effort.

**Supplementary Materials:** The following supporting information can be downloaded at: https://www.mdpi.com/article/10.3390/thalassrep12040019/s1, Table S1: Comparison of complication prevalence between two β-thalassaemia cohorts from Cyprus and Greece; Table S2: Comparison of serum ferritin concentration distribution in transfusion-dependent β-thalassaemia (TDT) patients between Cyprus and Egypt; Table S3: Comparison of cardiac T2* value and estimated liver iron concentration (LIC) in transfusion-dependent β-thalassaemia (TDT) population in different countries (NR, not reported); Table S4: Comparison of education level in β-thalassaemia patients in different countries

**Author Contributions:** Conceptualization, A.E., M.A.; methodology, M.A., D.F.; formal analysis, M.A., D.F., S.C., A.K.; data curation, S.C., A.K., I.S., E.P.; writing—original draft preparation, M.A.; writing—review and editing, D.F., A.E.; supervision, A.E. All authors have read and agreed to the published version of the manuscript.

**Funding:** This research received no external funding.

**Institutional Review Board Statement:** Ethical review and approval were waived for this study, due to the use of anonymized data from patient records for which patients have provided written authorization to their doctors for anonymized data analysis.

**Informed Consent Statement:** Informed consent was obtained from all subjects involved in the study.

**Data Availability Statement:** Data from medical records are available only to authorized treating physicians.

**Conflicts of Interest:** The authors declare no conflict of interest.

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
