# Peer review of "The Outcomes of Patients with Haemoglobin Disorders in Cyprus: A Joined Report of the Thalassaemia International Federation and the Nicosia and Paphos Thalassaemia Centres (State Health Services Organisation)"

_thalassrep, doi:10.3390/thalassrep12040019_

Round 1
Reviewer 1 Report
Extremely useful write up for workers in the filed of Thalassaemia.
Author Response
Our thanks to the reviewer
Reviewer 2 Report
This manuscript is a description of an impressive health care system provided to thalassemia patients in Cyprus, which can be a model for health care authorities and physicians treating those with hemoglobinopathies in other countries. Available outcomes are also generally well-described.
There are few comments and suggestions from this reviewer.
- Page 4, line 172-173: “Following processing and pre-storage filtration, packed red blood cell units are prepared for the transfusion of patients every 6-13 days on average…” . This sentence is not quite clear. Does it mean the processed blood will be transported to centers in need every 6-13 days (not everyday?).
- Page 4, line 175: It is mentioned that pre-transfusion Hb level is maintained at 10 g/dL. This reviewer think it might be worth to show the actual mean pre-transfusion Hb level achieved in the population in a relevant table (may be table 2?).
- Page 4, line 176: Is infectious screening of the blood products performed using nucleic acid test?
- Do all thalassemia patients receive treatment at one of the 4 large thalassemia centers of the country and may be in a few private hospitals? Are there any patients treating in regional hospitals? If so, it would be interesting to mention proportion of these patients.
- Pate 4, line 200: Dose the day care clinics also operate during weekend?
- Although this manuscript mainly focuses on non-transplanted thalassemia patients and their outcomes, the reviewer though it would be worth mentioning accessibility of BMT for thalassemia patients in the country. Who are eligible, who are not?
- The most impressive outcome to this reviewer is a very successful birth control of the new cases. In many South East Asia countries, where thalassemia is prevalence and the national policy for carrier screening at ante-natal care setting is also provide, does not achieve this level of control. It is worth sharing in a few sentences the key to success of affected birth control in Cyprus.
- Table 3, page 8: It is quite surprising that almost half of the patients (48%) had serum ferritin <1,000 ng/mL, given their mean age are 43 years. This is unlike other studies in many countries, including Egypt. In supplementary table 2, it was shown that, with mean age of patient 13 year, most patients had serum ferritin of >2,500 ng/mL. Could the authors make any comment or discussion on this issue? What are main chelators used in the country?

Author Response
We wish to thank the reviewer for the valuable comments and suggestions. All comments have been included as track changes in the word document. Only the pre-transfusion Hb table has not been made available (line 175). this has been requested from the clinic and may follow when received.
Reviewer 3 Report
The manuscript by Angastiniotis et al. describes the organization of healthcare services for haemoglobinopathies in Cyprus, with a particular focus on beta-thalassemia. They describe some parameters related to patient outcomes; some of which are clinical and related to patient survival, clinical complications and management and other are social in nature, related to marriage, education and employment. Overall this descriptive report is well written and will add to the similar reports in the literature already available from other hemoglobinopathy centers around the world discussing all these issues.
Some things for the authors to consider to improve the manuscript are as follows:
1. Table 1, column of β-thalassemia intermedia, last row: there seems to be a mistake in the sum? Please revise.
2. Section 5: Survival: “ There has been a 239 significant improvement in survival rates across the different birth cohort of patients”. It would be good to add a statement as to why survival rates are getting better and β-thalassemia patients are now living longer
3. Section 6: complications, when the authors discuss that malignancy is increasingly contributing to the causes of death in aged patients, it is recommended that the authors add few statements as to why we are seeing an increase in malignancy development and mention some of the factors that is leading to this despite the advancement in treatment modalities.
4. Section 6: Complications: Line 283: There exists a more recent reference that the one used by the authors on malignancy development in thalassemia (Blood Rev. 2019 Sep;37:100585. doi: 10.1016/j.blre.2019.06.002. Epub 2019 Jun 22.)
5. Section 7: Social aspects. There are two other publications on education, employment and marital status in β-thalassemia that the authors are encouraged to take into account:
- Pediatr Blood Cancer. 2010 Oct;55(4):678-83. doi: 10.1002/pbc.22565.
- Hemoglobin. 2020 Jul;44(4):278-283. doi: 10.1080/03630269.2020.1797776. Epub 2020 Jul 29.
6. Section 7: Social Aspects: What about psycho-social aspects? Did the authors look at the incidence of
Depression, Anxiety, and Stress Symptoms in in their cohort of patients? If not, authors are encouraged at least to mention few sentences and supporting references related to this at the end of that section and the importance of having more studies related to psycho-social aspects and counselling in the lives of these patients.
Author Response
Many thanks for these valuable comments. They have been included in the text as tracked changes.
Round 2
Reviewer 3 Report
no further comments